# 3′-Demethoxy-6-O-Demethylisoguaiacin and Norisoguaiacin Nematocidal Lignans from *Artemisia cina* against *Haemonchus contortus* Infective Larvae

**DOI:** 10.3390/plants12040820

**Published:** 2023-02-12

**Authors:** Rosa Isabel Higuera-Piedrahita, Mariana Dolores-Hernández, Héctor Alejandro de la Cruz-Cruz, Raquel López-Arellano, Pedro Mendoza-de Gives, Agustín Olmedo-Juárez, Jorge Alfredo Cuéllar-Ordaz, Manasés González-Cortazar, Ever A. Ble-González, María Eugenia López-Arellano, Alejandro Zamilpa

**Affiliations:** 1Facultad de Estudios Superiores Cuautitlán, Universidad Nacional Autónoma de México, Cuautitlán 54714, Mexico; 2Centro Nacional de Investigación Disciplinaria en Salud Animal e Inocuidad, Instituto Nacional de Investigaciones Forestales, Agrícolas y Pecuarias, Jiutepec 62574, Mexico; 3Centro de Investigación Biomédica del Sur, Instituto Mexicano del Seguro Social, Argentina No. 1, Centro, Xochitepec 62790, Mexico; 4División Académica de Ciencias Básicas, Universidad Juárez Autónoma de Tabasco, Carretera Cunduacán-Jalpa Km. 0.5, Cunduacán 86690, Mexico

**Keywords:** *Artemisia cina*, isoguaiacin, norisoguaiacin, lignans, *Haemonchus contortus*

## Abstract

*Artemisia cina* is a plant used in traditional Chinese medicine as a remedy for parasitic diseases. This study describes the isolation and chemical characterization of anthelmintic compounds of *A. cina* against *Haemonchus contortus* infective larvae (L_3_) through lethal testing. Previously, three extracts—*n*-hexane (HexAc), ethyl acetate (EtOAc) and methanol (MeOAc)—were evaluated at concentrations of 4 to 0.5 mg/mL, resulting in the HexAc extract with the greatest effect of 76.6% mortality of the larvae at 4 mg/mL. Then, this was chemically fractioned by polarity, obtaining seven fractions (C1F1–C1F7), and, when evaluated at concentrations from 2 to 0.25 mg/mL, the 2 mg/mL C1F5 fraction produced an effect against the nematode *H. contortus* of 100% mortality of the larvae. Thus, this fraction was fractionated again by column chromatography, obtaining twelve subfractions (C2F1–C2F12) which were evaluated from 1 to 0.125 mg/mL, with the C2F5 subfraction causing a nematicidal effect of 100% mortality. NMR analysis of one (^1^H, ^13^C and DEPT) and two dimensions (COSY, HSQC and HMBC) and mass spectrometry of this fraction allowed us to identify the mixture of 3′-demethoxy-6-O-demethylisoguaiacin and norisoguaiacin. Therefore, it can be assumed that the mixture of these compounds is responsible for the anthelmintic effect. These results indicate that *A. cina* containing anthelmintic compounds and might be used as an antiparasitic drug against *H. contortus*.

## 1. Introduction

*Haemonchus contortus* is one of the most pathogenic gastrointestinal nematodes (GINs) that causes inflammation and micro-hemorrhages in abomasum, and it is also responsible for important economic losses in flocks worldwide [1]. This parasite evades the host immune response and has developed mechanisms against anthelmintic commercial drugs [2]. Commercial anthelmintic molecules are the main strategies used for the control of *H. contortus* [3]. However, resistance has been growing and the commercial molecules have reduced efficacy [3]. Control may constitute an integrated parasite method such as herbolary [4]. Some plants have been used in traditional medicine as potential anthelmintics, such as the Asteraceae family [5]. The antiparasitic compounds from the Asteraceae family are described as sesquiterpenes, lactones, phenols as tannins or gallic acid [6]. The anthelmintic compounds described below are reported as members of the genus *Artemisia* [7]. *Artemisia* was used against *Trypanosoma cruzi* and *Giardia lamblia* infections in humans [8].

The family Asteraceae (Compositae) is present on all continents and has significance as horticultural and garden ornaments as well as medicinal plants, vegetables and pesticides. The most reported molecule of this family is artemisinin from *Artemisia annua*, which has a lethal effect on multi-resistant *Plasmodium falciparum* [9].

The most important molecules of the Asteraceae family are the sesquiterpene lactones, as dehydrozaluzanin C and brevilin A from *Munnozia maronii* and *Centipeda minimum*, respectively, are both highly effective antiparasitic compounds against *Plasmodium*, *Leishmania*, and *Trypanosoma*. For example, wild chimpanzees chewing stems of *Vernonia amygdalina* (Asteraceae) led to the development of the antiplasmodial sesquiterpenes vernodalin, vernolide, and hydroxyverniladin [9].

Other compounds are reported in this family, such as phenolic compounds. In-vitro Trypanosoma cruzi trypomastigotes cannot proliferate in the presence of gallic acid and its derivatives [10]^.^ Flavonoids are the component class with the highest levels in terms of prevalence and chemical stability. Flavones and flavonols are present in all Asteraceae; metabolites as apigenin, luteolin, kaempferol, quercetin; and the derivatives of the flavanone (-)-epigallocatechin and -epicatechin [9].

The plant *Artemisia cina* has been used as an aqueous extract to treat helminth infections [11]. In 1950, *A. cina* was used as an anthelmintic in children infected with *Ascaris lumbricoides* at 60 to 600 mg/day for three consecutive days [12]. A santonin molecule was recently identified as an anthelmintic compound from *A. cina* [12]. Santonin may cause toxicity in high doses and affect the central nervous system [12].

Klayman (1975) identified artemisinin as a majoritarian compound and proposed it as an antimalarial drug [13]. Artemisinin was found in leaves, and it was used to treat soldiers with malaria [13]. In addition, Bashtar et al. reported the impact of *A. cina* on the sheep cestode *Moniezia expansa* 14 h post-treatment [11]. *A. cina* ethanolic extract also showed anthelmintic activity in ruminants by sesquiterpenes and lignans [14]. Antiparasitic activity was reported by Higuera-Piedrahita against *H. contortus* adults at 4 mg/mL with 86.9% mortality after six hours post-exposition [15]. Likewise, periparturient goats were treated with an *n*-hexane extract of *A. cina* and an egg-per-gram reduction was found at seven and twenty-three days after oral treatment [15]. *A. cina* showed an anthelmintic effect against *H. contortus* and *Teladorsagia circumcincta* [15]. The ethanolic extract of *A. cina* was administered to young lambs infected with *H. contortus* and a reduction by 63.2% in the egg per gram was found [14].

*Artemisia cina* extract showed upregulation of the *H. contortus Hc29* gene (associated with glutathione peroxidase as a toxicity mechanism for parasites) by 13 fold (*p* < 0.01) on transitional larvae (from xL_3_ to L_4_, first endoparasite stage) at 0.078 mg/mL [14]. An *n*-hexane extract of this plant showed a 100% reduction in gerbils (*Meriones unguiculatus*) artificially infected with *H. contortus* infective larvae [14].

Other molecules responsible for *A. cina* anthelmintic activity may be under study to improve the nematode control and contribute to reducing the anthelmintic-resistance problem [16]. Assessing the nematocidal activity of plant extracts and metabolites provides tools for integrated parasitic control [17]. The aims of this study were to identify the most active extract (*n*-hexane, ethyl acetate and methanol) from *A. cina* against the egg and infective larval stages of *H. contortus* and to evaluate the lethal effect of different fractions obtained from the *n*-hexane extract of *A. cina* considered in a previous study [14], to identify anthelmintic compounds from the *n*-hexane *A. cina* extract against *H. contortus* infective larvae.

## 2. Results

### 2.1. Larvicidal Activity of Artemisia cina Extracts

The nematocidal activity of *n*-hexane, ethyl acetate and methanol extracts against *H. contortus* eggs and larvae are shown in Table 1. The best activity was founded in *n*-hexane extract of *A. cina* against larvae and eggs.

### 2.2. Larvicidal Activity of Fractions of n-Hexane Artemisia cina Extract

Larvicidal activity was found in the fraction C1F5 with 100% mortality of *H. contortus* L_3_. In addition, the fraction exhibited dose-dependent activity (Table 2). The lethal concentration of 90 (LC_90_) was 1.26 mg/mL.

### 2.3. Larvicidal Activity of Compounds Obtained of n-Hexane Artemisia cina Fractioning

The best anthelmintic activity was found for the subfraction named C2F5 with 100% at 1 mg/mL against *H. contortus* L_3_ (Table 3). The LC_90_ was found at 0.7 mg/mL. The in-vitro assay of the C2F5 subfraction showed the dose-dependent activity of the C2F5 subfraction.

The C2F5 subfraction had the highest performance compared to the other molecules; therefore, the fraction was identified using nuclear magnetic resonance (NMR).

### 2.4. Identification of 3′-Demethoxy-6-O-Demethylisoguaiacin and Norisoguaiacin

The analysis of the NMR spectra of one and two dimensions (Figure A1, Figure A2, Figure A3, Figure A4, Figure A5 and Figure A6) and the comparison of the spectroscopic data (Table 4) with those described in the literature [18] allowed the identification of the mixture of two lignans known as 3′-demethoxy-6-*O*-demethylisoguaiacin (**1**) and norisoguaiacin (**2**). The chemical structures are showed in Figure 1.

Mass was confirmed by the mass spectrometry of compound 1: 309 m/z [Mass + Na + 2H^+^] and norisoguaiacin 315 m/z [Mass + Na + 2H^+^] (Figure A7).

The experiments showed that compound one (**1**) is present in the mixture at a proportion of 63%; thus, the other compound constitutes 37%. We found that the anthelmintic effect found would be caused by the cooperation of the two lignans of the mixture.

## 3. Discussion

The Asteraceae family is considered capable of important biological activities such as anti-inflammatory, anticancer, antitumor, hepato-protective, antimicrobial, antiparasitic, CNS depressant and antioxidant behavior [19]. *A. cina*, also named wormseed, can be proposed as an anthelmintic alternative in the strategy of parasite integrated control. Lignans dilucidated in the present study represent molecules with potent anthelmintic activity. In addition, *A. cina* aqueous extract was reported by Bashtar et al. [11] as an anthelmintic tool against *M. expansa* after 3 h of exposure. Lans et al. [20], in 2007, reported that *A. cina* had an antiparasitic effect against roundworms and pinworms and amoebal infections [20]. The present study shows the anthelmintic effect of HexAc extract and molecules (isoguaiacin and norisoguaiacin) against *H. contortus*, with L_3_ showing 100% effectiveness.

Plants with anthelmintic activity typically contain saponins, alkaloids, amino-acids, tannins, polyphenols, lignans, glycosides, lactones, terpenes and phenolic compounds [21]. The present study shows two lignan mixtures combine to enable the anthelmintic activity. The active compounds are plant secondary-metabolite products, which have been associated with defensive mechanism against herbivore grazing [22]. For example, walnut contains naphthoquinone, which is the active compound against worms, and glycosides are active against cestodes in goats [22].

*Artemisia* spp. is abundant in fatty acids, phenolic acids, coumarins, isocoumarins, flavonoids, and terpens. The leaves, steams, and flowers have active metabolites which can be used infused, decocted or minced and used as anthelmintic, antidiabetic, antihypertensive, emmenagogue, and antivenom methods to treat digestive and cutaneous problems [23].

Ruminant production is threatened by sustainability issues. At the same time, the anthelmintic resistance, and the risk of antiparasitic residues in the environment, is growing. The small ruminant sector should adopt sustainable practices in order to become more resilient, environmentally friendly and productive [23]. Sustainability represents the future of global agriculture and livestock. Therefore, the increase in resistant populations, the drug residues in meat and milk, and the high costs of treatments require sustainable solutions for helminth control and also a control which includes not only anthelmintics [23]. Control can take the form of alternatives such as natural compounds, plant extracts or molecules related to ethnoveterinary. The evidence of the anthelmintic properties of plants suggests a promising alternative for parasite control. The use of ethnoveterinary plant extracts has been reported around the world. Plants are an alternative for treatment of organic livestock [23].

Some compounds with anthelmintic activity from plant extracts have been reported, such as methyl gallate, condensed tannins and gallic acid from *Caesalpinia coriaria* fruits and leaves, the authors reported a concentration-dependent ovicidal effect of hydro-alcoholic extracts from both leaves and fruits. The EC50 from fruit and leaves extracts was established as 1.63 and 3.91 mg/mL, and as 3.98 and 11.68 mg/mL, against *H. contortus* and *H. placei*, respectively [24]. Mravčáková et al. (2020) [25] reported phenolic compounds, flavonoids and phenolic acids from wormwood (*Artemisia absinthium* L). In addition, the authors reported flavonoids with antioxidant properties such as apigenin and luteolin, flavonols such as kaempferol and quercetin and flavanones such as naringenin, which may also perform anthelmintic activity [25]. In the present study, we did not find the latter compounds; the anthelmintic properties were those of 3′-demethoxy-6-O-demethylisoguaiacin and norisoguaiacin, a mixture of lignans.

*Artemisia absinthium* Linn. reported a significant anthelmintic effect on live adults of *H. contortus.* The extract induced the paralysis or worms’ death post-treatment [26]. Iqbal et al. (2004) [27] found alcoholic extracts of *Artemisia brevifolia* cause mortality of *H. contortus*. In the present study, the higher mortality of infective larvae was caused by the *n*-hexane extract of *A. cina.*

*Artemisia vestita* and *A. maritima* methanol extracts have significant anthelmintic activity against infective larvae and adults of *H. contortus.* Both plants showed an anthelmintic effect at a lower concentration from leaves and stems. The activity of the crude aqueous extract of *A. maritima* against different GINs stages of sheep make it an important component of the diet of animals [28].

*Artemisia brevifolia* reported anthelmintic activity against *H. contortus*, as evident from the mortality of the worms. The crude aqueous extract of *A. brevifolia* paralyzes the *H. contortus* adult, but when the parasite is washed with PBS for 30 min the adult recovers the motility. The same plant exhibited an antiparasitic effect against Strongyloides [29].

Some molecules responsible for an anthelmintic effect are terpenoids. Terpenoids are reported from *A. lancea*, wherein 63.14% are oxygenated monoterpene, 10.93% sesquiterpene hydrocarbon, 10.21% oxygenated sesquiterpenoid and 6.70% monoterpene hydrocarbon [30]. Terpenoids perform insecticidal activity such as the inhibition or retardation of growth, maturation damage, reduced reproductive capacity, and appetite suppression. *A. lancea* has 1,8-cineole and camphor, which belong to the terpenoid class [30]. The author reported that the presence of terpenoids induces an anthelmintic property, but the synergism with minoritarian compounds increases antiparasitic activity. The minor constituents such as terpinen-4-ol, spathulenol, caryophyllene oxide, caryophyllene, copaene, geraniol, borneol, and isocineole are present in *A. lancea* [30]. In the present study, the compounds that perform activity are lignans and the synergistic activity of minor compounds may be probed. In addition, 3′-demethoxy-6-O-demethylisoguaiacin and norisoguaiacin have an additive action against infective larvae.

The compound described as isoguaiacin is also called (5R,6R,7R)-5-(4-hydroxyphenyl)-6,7-dimethyl-5,6,7,8-tetrahydronaphthalene-2,3-diol and its common name is 3′-demethoxy-6-*O*-demethylisoguaiacin. The molecule was found at a proportion of 63% in the lignan mixture. Isoguaiacin was patented in 1989, number CAS:71113-16-1. Konno et al. [18] found the lignan from the phytochemical analysis of *L. tridentata*. Konno et al. [31] described the chemical structure of isoguaiacin as ^1^H NMR [31]. Furthermore, lignans have a broad spectrum of action as secondary metabolites produced under plant-stress and for self-defense purposes [32]. For instance, lignans are important as anti-fungals and insecticidals and against other kinds of parasites [33], and against microorganisms such as virus and bacteria [34]. Gnabre et al. [35] reported antiviral activity in humans using isoguaiacin, using the extract from *L. tridentata*.

Previous studies mentioned the isoguaiacin efflux on the bacteria cell membrane to inhibit the ATP binding (from the ABC box-transporters) inducing bacteria death [36]. The possible mechanisms of isoguaiacin induces vascular muscle relaxation in rodents [36]. In addition, isoguaiacin is considered an antihypertensive drug [11,36]. The possible mechanisms of isoguaiacin action include being antihypertensive for the participation of the nitrous oxide (NO) pathway, expression of second messengers such as cyclic guanosine monophosphate (cGMP) and alteration of potassium channels sensitive to H2S and ATP in the physiological cascade of vasodilation [37]. Another mechanism of action is related to the endothelium-independent vasodilatation pathway [37].

On the other hand, the lignan norisoguaiacin had an effect on the inhibition of the mitochondrial electrons related to the transport system, where the lignan was found in 37% of the mixture dilucidated obtained from the plant *Larrea divaricata* [38]. In addition, norisoguaiacin mechanisms are related to the reactive oxygen species (ROS) and in the ATP reaction. For instance, inhibition of the enzyme NADH oxidase on bovine heart mitochondria [38]. Gisvold and Thaker [39] also showed that norisoguaiacin inhibits the enzyme formyltetrahydrofolate synthetase and carboxylesterase. In addition, the compound inhibits phagocytosis and adheres to various binding sites, which prevents important metabolic reactions at the mitochondrial level [39].

Norisoguaiacin has antiviral properties against HIV [35]. Norisoguaiacin and meso-nor-dihydroguaiaretic acid obtained from *Larrea nitida* have antioxidant activity demonstrated with ABS transporters which derive cationic radicals [40]. Norisoguaiacin has anti-protozoal properties against *Trypanosoma brucei rhodesiense*, *Trypanosoma cruzi*, *Leishmania donovani* and *Plasmodium falciparum.* The lignan caused cytotoxic effects on rat myoblasts at doses of 25.4 µg/mL, therapeutic doses (LD_50_) were 2.8, 14.6, 5.2, 2.9 ug/mL, respectively [41] Furthermore, norisoguaiacin was reported in 2014 by Song et al. [42] as a metabolite which acts on female hormones; thus, it was also considered a phytoestrogen, which has practical applicability in the prevention of breast and uterine cancer, in addition to reducing menopause symptoms. Norisoguaiacin is proposed as a supplement in the human diet through functional foods to prevent female cancer and menopausal symptoms [42].

In the present study, we showed isoguaiacin and norisoguaiacin are potent anthelmintics; although the mechanism of action is still unknown, lignans induce death against infective larvae of *H. contortus* at 24 h post-exposure. One possible mechanism might be the activity on the ROS system as it was observed in a previous study with *H. contortus* and *A. cina n*-hexane (Higuera-Piedrahita et al., 2019). However, more studies are required to confirm these mechanisms.

## 4. Materials and Methods

### 4.1. Plant Material

The fresh pre-flowering leaves and stems of *A. cina* O. Berg ex Poljakov (Asteraceae) (10 kg) were bought at Hunab laboratory. A voucher specimen was authenticated by Dr. Alejandro Torres-Montúfar and was deposited at the herbarium of Facultad de Estudios Superiores Cuautitlán (FES-C) UNAM, México under voucher no 11967. The plant was grown at 80% humidity, 24 °C temperature and pH = 6.3 soil.

### 4.2. Artemisia cina Extract Obtaining and Chemical Fractioning

Dry *A. cina* leaves and stems (10 kg) were ground and placed in 46 L crystal containers. Extraction using leaves and stems was performed using *n*-hexane, ethyl acetate and methanol maintained for 24 h at room temperature (23–25 °C). Extracts were filtered using a Whatman No. 4 paper and the solvent were removed by low-pressure distillation using a rotary evaporator (Heidolph Laborota 4000, Heidolph Instruments, Schwabach, Germany) which was finally lyophilized. The extracts were kept at 4 °C for phytochemical and biological assays. The in-vitro anthelmintic activity was evaluated using *H. contortus* larvae as biological model. The extract with the most anthelmintic effect was selected and evaluated using bio-guided assays.

### 4.3. Isolation and Identification of 3′-Demethoxy-6-O-Demethylisoguaiacin (1) and Norisoguaiacin (2) from the A. cina n-Hexane Extract

The *n*-hexane extract was processed using chromatographic techniques with silica gel in an open column (200 g, 70–230 mesh; Merck, Darmstadt, Germany). The extract was eluted using a gradient system with *n*-hexane and ethyl acetate, with growing polarity as mobile phase starting with 100% *n*-hexane and ending with 100% ethyl acetate and 187 fractions were obtained. Fractions were grouped depending on their chemical similarity and monitored using thin-layer chromatography and concentrated using a rotary evaporator.

The obtained fractions were C1F1 (4 g), C1F2 (3.1 g), C1F3 (2.1 g), C1F4 (5 g), C1F5 (2.5 g) and C1F6 (3.2 g). All fractions were assessed against *H. contortus* larvae. The most active fraction (C1F5) was separated using chromatographic fractionation in a glass column with silica gel (200 g, 70–230 mesh; Merck, Darmstadt, Germany). The mobile phase was *n*-hexane and ethyl acetate and 12 fractions were obtained. The fractions were grouped according to thin-layer chromatography analysis, resulting in: C2F1, C2F2, C2F3, C2F4, C2F5, C2F6, C2F7, C2F8, C2F9, C2F10, C2F11, C2F12. The most bioactive subfraction C2F5 was selected and dilucidated as an anthelmintic molecule. Compound identification was elucidated using ^1^H and ^13^C NMR analysis, using Agilent DD2-600 spectrometer with one NMR probe at 25 °C in CD_3_OD and DMSO-D6 (Cambridge Isotope Laboratories Inc., Tewksbury, MA, USA) as a solvent and TMS as reference.

### 4.4. Mass Spectrometry Analysis of Compounds

Molecular weights of the isolated compounds were determined by mass spectrometry using a triple/quadruple mass spectrometer (TQ/MS; ACQUITY Ultraperformance coupled to an MS Xevo electrospray source, Waters Corp. Milford, MA, USA). This system was operated using the positive electrospray ionization mode (ESI+) with methanol HPLC grade as solvent.

### 4.5. Haemonchus contortus Larvae Obtaining

The *H. contortus* larvae were obtained from artificially infected lamb (FESC strain). Four-month old, parasite free, and with 30 kg body weight, male lamb was orally infected with 5000 infective larvae. After a 20-day, pre-patent period, faeces were collected directly from the rectum of the animal. The infective larvae of the parasite were obtained from faecal cultures using the modified Corticelli–Lay technique. After 14-day incubation period at 24 °C, infective larvae were recovered. The egg-donor lamb was maintained under controlled conditions according to principles of animal welfare and the elimination of unnecessary animal suffering accord to Norma Oficial Mexicana DOF 07-06-2012.

### 4.6. Larval Mortality Assay

The assays were carried out in 96-well microtiter plates (three wells per treatment). The experiment was performed in triplicates. Treatments were established according to phytochemical separation. Fractions C1F1-C1F7 were tested at 2, 1, 0.5 and 0.25 mg/mL. Fractions C2F1 to C2F12 were tested at 1, 0.5, 0.25 and 0.125 mg/mL. Each assay was assessed using negative control (distilled water) and positive control (Ivermectin 0.5 mg/mL). *H. contortus* larvae (100 L_3_ per well) were added to microtiter plates and treatment was added. The microtiter plates were incubated at room temperature for 24 h (n = 6).

After incubation, 20 aliquots of 2 µL were removed from each well and deposited into a slide for microscopic examination. Dead and alive larvae were counted and the mortality percentage was calculated using the following formula:$$\%\mathrm{Mortality}=\frac{dead\;larvae\;mean}{alive\;larvae\;mean+dead\;larvae\;mean}\times 100$$

### 4.7. Statistical Analysis

Larval mortality percentages were normalized using a square root transformation a random and analysed through an ANOVA on a completely random design. Differences among means were compared using the Tukey test (*p* < 0.05). Lethal concentrations (LC_50_ and LC_90_) were determined with the PROBIT procedure included within the SAS statistic package.

## 5. Conclusions

This study reveals that *A. cina* is a potential source of promising compounds with a range of useful bioactivities that support its use in traditional medicine. The bio-guided study allows the identification of the compound(s) with anthelmintic activity. The mixture of lignans (3′-demethoxy-6-O-demethylisoguaiacin and norisoguaiacin) are the compounds responsible for the in-vitro anthelmintic activity against the eggs and infective larvae of *H. contortus*. Therefore, *A. cina* can be proposed as a candidate for a nematicidal phytopharmaceutical.

## Figures and Tables

**Figure 1 plants-12-00820-f001:**
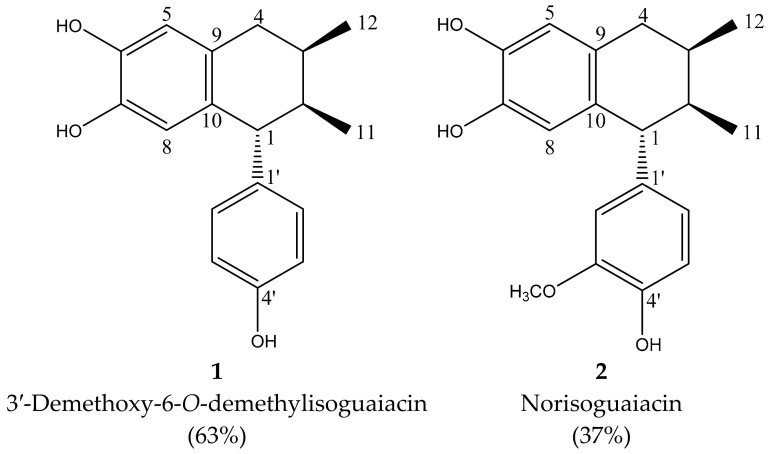
Chemical structure of 3′-demethoxy-6-O-demethylisoguaiacin (1) and norisoguaiacin (2).

**Table 1 plants-12-00820-t001:** In-vitro lethal activity of *n*-hexane, ethyl acetate and methanol extract of *A. cina* against *H. contortus* infective larvae and eggs after 24 h and 48 h, respectively.

*A. cina* Extract (mg/mL)	Egg Hatching Inhibition after 48 h Post-Exposition (EHI% ± SD)	Infective Larvae Mortality after 24 h Post-Exposition (Mortality% ± SD)
*n*-hexane extract		
4	76.6 ± 9.3 ^a^	80 ± 9 ^a^
2	63.6 ± 5.9 ^a^	68 ± 7 ^a^
Ethyl acetate extract		
4	61.1 ± 3.4 ^b^	45 ± 5 ^b^
2	56.8 ± 9.3 ^b^	20 ± 2 ^b^
Methanolic extract		
4	39.6 ± 1.2 ^c^	30 ± 3 ^c^
2	10.4 ± 3 ^c^	10 ± 1 ^c^
Ivermectin (5 mg/mL)	100 ^d^	100 ^d^

Means with different letters (^a,b,c,d^) in same column indicate statistical differences (*p* < 0.05). SD: standard deviation.

**Table 2 plants-12-00820-t002:** In-vitro mortality percentages attributed to the effect of seven *n*-hexane *A. cina* fractions against *H. contortus* infective larvae.

Fractions on *n*-Hexane Extract of *A. cina*	Concentration (mg/mL)	% Mortality	LC_50_/LC_90_ (mg/mL)
C1F1	2	0	-
	1	0	-
	0.5	0	-
	0.25	0	-
C1F2	2	0	-
	1	0	-
	0.5	0	-
	0.25	0	-
C1F3	2	0	-
	1	9.11 ± 8.8 ^d^	-
	0.5	11.24 ± 20 ^d^	-
	0.25	0	-
C1F4	2	9.09 ± 0.8 ^d^	-
	1	0	-
	0.5	0	-
	0.25	3.8 ± 1.2 ^e^	-
C1F5	2	100 ^a^	0.63/1.26
	1	94.97 ± 5 ^b^	-
	0.5	35.87 ± 14 ^c^	-
	0.25	9.43 ± 3.7 ^d^	-
C1F6	2	69.46 ± 5.7 ^b^	0.6/5.5
	1	67.44 ± 15.8 ^b^	-
	0.5	36.84 ± 3.6 ^c^	-
	0.25	26.69 ± 4.1 ^c^	-
C1F7	2	8.51 ± 0.5^d^	4.15/14.9
	1	8.81 ± 4.9^d^	-
	0.5	1.92 ± 1.6 ^e^	-
	0.25	4.13 ± 0.3 ^e^	-

Means with different letters ^(a,b,c,d,e)^ in same column indicate statistical differences (*p* < 0.05). SD: standard deviation.

**Table 3 plants-12-00820-t003:** Mortality percentage and lethal-effect concentration (LC_50_ and LC_90_) of compounds obtained from *A. cina n*-hexane extract C1F5 fraction after 24 h of exposition.

Fractions on *n*-Hexane Extract of *A. cina*	Concentration (mg/mL)	Mortality %	LC_50_/LC_90_ (mg/mL)
C2F1	2	0	-
	1	0	-
	0.5	0	-
	0.25	0	-
C2F2	2	0	-
	1	0	-
	0.5	0	-
	0.25	0	-
C2F3	2	0	-
	1	9.11 ± 8.8 ^d^	
	0.5	11.24 ± 20 ^d^	
	0.25	0	
C2F4	2	9.09 ± 0.8 ^d^	-
	1	0	
	0.5	0	
	0.25	3.8 ± 1.2 ^e^	
C2F5	2	100 ^a^	0.63/1.55
	1	94.97 ± 5 ^b^	
	0.5	35.87 ± 14 ^c^	
	0.25	9.43 ± 3.7 ^d^	
C2F6	2	69.46 ± 5.7 ^b^	0.6/5.5
	1	67.44 ± 15.8 ^b^	
	0.5	36.84 ± 3.6 ^c^	
	0.25	26.69 ± 4.1 ^c^	
C2F7	2	8.51 ± 0.5 ^d^	-
	1	8.81 ± 4.9 ^d^	
	0.5	1.92 ± 1.6 ^e^	
	0.25	4.13 ± 0.3 ^e^	
C2F8	2	18.3 ± 1.5 ^d^	-
	1	10.3 ± 1.5 ^d^	
	0.5	0	
	0.25	0	
C2F9	2	46 ± 3.6 ^c^	-
	1	15 ± 5 ^d^	
	0.5	0	
	0.25	0	
C2F10	2	23.3 ± 9.8 ^d^	-
	1	14.3 ± 4.04 ^d^	
	0.5	0	
	0.25	0	
C2F11	2	100 ^a^	0.25/0.65
	1	100 ^a^	
	0.5	98.33 ± 1.5 ^a^	
	0.25	87 ± 1.2 ^b^	
C2F12	2	91.7 ± 1.5 ^b^	-
	1	91 ± 1 ^b^	
	0.5	53.67 ± 3 ^d^	
	0.25	10 ^d^	
Distillated water	-	0	-
Ivermectin	5	100 ^a^	-

Means with different letters (^a,b,c,d^) in same column indicate statistical differences (*p* < 0.05). SD: standard deviation.

**Table 4 plants-12-00820-t004:** Spectral data of ^1^H-NMR (600 MHz) and ^13^C-NMR (150 MHz) 3′-Demethoxy-6-O-demethylisoguaiacin (**1**) and norisoguaiacin (**2**) in CD_3_OD and chemical structures of molecules.

Position	1 δ _H_ (J in Hz)	1 δ_C_	2 δ _H_ (J in Hz)	2 δ_C_
1	3.53 (1H, d, 6.2)	51.1	3.53 (1H, d, 5.1)	51.5
2	1.86 (1H, d, 2.5, 6.6)	42.2	1.91 (m)	42.0
3	1.98 (m)	30.5	1.98 (m)	30.8
4 a b	2.80 (1H, dd, 5.1, 16.1) 2.37 (1H, dd, 6.6, 16.1)	35.9	2.83 (1H, dd, 5.1, 12.8) 2.38 (1H, dd, 4.7, 16.1)	36.0
5	6.51, s	116.0	6.51 (1H, br, s)	116.0
6		144.5		144.4
7		144.1		144.4
8	6.21, s	118.0	6.24, (1H, s)	117.9
9		130.8		130.7
10		128.5		128.5
11	0.86 (3H, d, 6.2)	16.1	0.88 (3H, d, 5.1)	16.0
12	0.87 (3H, d, 6.2)	16.1	0.86 (3H, d, 6.2)	16.2
1′		139.7		140.4
2′	6.82 (1H, d, 8.4)	130.9	6.57 (1H, br, s)	113.5
3′	6.66 (1H, d, 8.4)	115.6		148.6
4′		156.1		145.3
5′	6.66 (1H, d, 8.4)	115.6	6.68 (1H, d, 7.7)	115.5
6′	6.82 (1H, d, 8.4)	130.9	6.45 (1H, d, br, 7.7)	122.8
OCH_3_	----	----	3.73 (3H, s)	56.2

## Data Availability

Data is unavailable due to privacy.

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
