# Peer review of "3′-Demethoxy-6-O-Demethylisoguaiacin and Norisoguaiacin Nematocidal Lignans from *Artemisia cina* against *Haemonchus contortus* Infective Larvae"

_plants, 2023, doi:10.3390/plants12040820_

Round 1
Reviewer 1 Report
The manuscript of Rosa Isabe Higuera-Piedrahita et al. “3'-demethoxy-6-O-demethylisoguaiacin and norisoguaiacin nematocidal lignans from Artemisia cina against Haemonchus contortus infective larvae” is devoted to the study of the activity of Artemisia cina extracts against Haemonchus contortus larvae. The authors obtained extracts from leaves and stems of Artemisia cina in 3 different solvents: n-hexane, ethyl acetate and methanol. The extract with the greatest effect (HexAc) was then separated into fractions using a chromatography column. A faction characterized by 100 % mortality of the Haemonchus contortus larvae was analyzed by spectroscopic methods to determine its composition.
In general, the manuscript presents interesting data, is well organized and documented, and written in acceptable form.
However, before publishing, it is necessary to make corrections:
1) Table 1
The way the results are presented is unclear to me. I understand that for each extract the tests were performed at two different concentrations: 4 and 2 mg/mL. The title of the table states that the tests were conducted after 24 and 48 hours. Which results are included in the table? It is not specified. There is also no legend - what do the letters a, b, c, d mean? – Sentence: “Means with different letter into same column indicate statistic differences (p<0.05)” is unclear to me. This also applies to tables 2 and 3.
2) Tables 2 i 3 – last columns
Firstly, the description in both tables should be unified, i.e. consistently use either the full name or the abbreviation of the parameter. In addition, the way the data is presented in this column suggests a range of values, not two different designated parameters. Therefore, I would suggest writing LC50/LC90 in the column header and the values according to this scheme in the following rows, e.g. 0.63/1.55.
3) Tables 1-3: bold or highlight the best results in a different color. It will be more readable.
4) Table 4
Some of the δ values given in the table do not coincide with the values shown in the NMR spectra. I understand it's a matter of reading accuracy. However, I would suggest analyzing these values and unifying them, especially when the signals are singlets.
5) Discussion:
Line144 – The wormseed wallflower (Erysimum cheiranthoides L.) belongs to Brassicaceae family, not to Asteraceae family. Placement of the fragment: „The wormseed wallflower was used for deworming pigs and dogs with tincture and aqueous extract…” here is not appropriate.
Line 145: I think it should be santonin instead of santonica
Line 156: „The compound described as isoguaiacin is also called…” – This is not correct. Isoguaiacin is: (6R,7R,8R)-8-(4-hydroxy-3-methoxyphenyl)-3-methoxy-6,7-dimethyl-5,6,7,8-tetrahydronaphthalen-2-ol, CAS: 78341-26-1 (see PubChem). The described compound is a derivative of the isoguaiacin. Its correct systematic name is: (5R,6R,7R)-5-(4-hydroxyphenyl)-6,7-dimethyl-5,6,7,8-tetrahydronaphthalene-2,3-diol. This should be corrected throughout the manuscript.
Author Response
Authors are very grateful for your observations. Responses are included in the attached PDF

Reviewer 2 Report
In the manuscript “3'-demethoxy-6-O-demethylisoguaiacin and norisoguaiacin nematocidal lignans from Artemisia cina against Haemonchus contortus infective larvae” the authors describe the bioassay-guided isolation of active fractions of Artemisia cina. The authors identified two known lignans and determined the activity of the mixture. The results are interesting, but the manuscript needs to be revised and proofread. The following are general comments about the manuscript:
# How were the extractions made, serial or parallel? How much time in each solvent and how much crude extract was obtained for each solvent?
# The two compounds were obtained in mixtures; in some parts of the manuscript, it is not clear. Also, the identification of both is from a mixture, not a pure sample.
# How is the activity of the mixture compared to the individual compounds? Other lignans?
The following are specific comments:
# Legend for table 1 is confusing.
# Line 95, grammar issues.
# Line 104 and 106, “… compound named C2F5…”, C2F5 is not a compound, it is a fraction based on what the authors described.
# The authors should carefully refer to the fractions consistently, C2F5 has been defined as compound, fraction, and subfraction.
# Line 125. The authors claimed that the action could be a cooperation of the two lignans found in the mixture. This is a very strong statement, they need to reference it and further discuss it with some evidence.
# Line 158, the authors mentioned that the proportion isoguaiacin is 66%, while in line 124 is 63%.
# In lines 163-170 the authors discuss other anti-infective activities unrelated to the antihelminthic activity of the lignans, it doesn’t belong to the discussion.
# The lines 174-177 need to be revised, the grammar is confusing and there is a reference missing.
# The sentence starting on line 207 makes no sense, there are typos and grammar issues.
# Line 215, the authors need to elaborate on the Garcia et al studies and how it relates or doesn’t relate to the results presented in this manuscript.
# Line 243, should be “monitored”.
# Line 251, should it be “elucidated”?
# Line 273, the subtitle needs to be revised. May be eliminated “by Bio-guided assay”.
# Line 275, revised the prepositions, it should be “in triplicates”.
# Line 276, “probed” is not the right verb.
# The examples of grammar issues and typos are not all-inclusive, there are many more, too many to list here.
Author Response
Authors are very grateful for you observations.
Responses are included in the attached PDF

Reviewer 3 Report
This study entitled “3'-demethoxy-6-O-demethylisoguaiacin and norisoguaiacin nematocidal lignans from Artemisia cina against Haemonchus contortus infective larvae” used chromatographic methods to screen out the bioactive compounds of A. cina against H. contortus. The results showed that C2F5 from the n-hexane extract exhibited the best nematicidal effect among all sub-fractions. Finally, two bioactive compounds were identified responsible for killing H. contortus. Some major points are shown below:
1. Since 3'-demethoxy-6-O-demethylisoguaiacin and norisoguaiacin were identified in this study, how about their contents in Artemisia cina. If the levels of these two compounds are quite low in Artemisia cina, their applications as antiparasitic agent may be limited, because preparation of these two compounds in large scale will be time consuming.
2. Line 226, 10 “kg”
3. Line 113, please provide the full name of RMN.
4. Line 104, why only C2F5 was selected as the highest performance compound? the mortality of C2F11 was also 100%
5. Line 122, Figure 7 is missing. Please provide it in the manuscript.
6. Line 328, Figure A7 is missing, too.
Author Response

(The authors gave the same response as above.)

Round 2
Reviewer 2 Report
All my concerns have been addressed by the authors.
Reviewer 3 Report
All questions have been answered and the manuscript has been revised according to the comments.